# KnowFi: Knowledge Extraction from Long Fictional Texts

**Cuong Xuan Chu**                                               CXCHU@MPI-INF.MPG.DE
**Simon Razniewski**                                          SRAZNIEW@MPI-INF.MPG.DE
**Gerhard Weikum**                                               WEIKUM@MPI-INF.MPG.DE
*Max Planck Institute for Informatics*
*Saarbrücken, Germany*

## Abstract

Knowledge base construction has recently been extended to fictional domains like multi-volume novels and TV/movie series, aiming to support explorative queries for fans and sub-culture studies by humanities researchers. This task involves the extraction of relations between entities. State-of-the-art methods are geared for short input texts and basic relations, but fictional domains require tapping very long texts and need to cope with non-standard relations where distant supervision becomes sparse. This work addresses these challenges by a novel method, called KnowFi, that combines BERT-enhanced neural learning with judicious selection and aggregation of text passages. Experiments with several fictional domains demonstrate the gains that KnowFi achieves over the best prior methods for neural relation extraction.

## 1. Introduction

**Motivation and Problem:** Relation extraction (RE) from web contents is a key task for the automatic construction of knowledge bases (KB). It involves detecting a pair of entities in a text document and inferring if a certain relation (predicate) holds between them. Extracted triples of the form (subject, predicate, object) are used for populating and growing the KB. Besides this major use case, RE also serves other applications like text annotation and summarization, semantic search, and more.

Work on KB construction has mostly focused on general-purpose encyclopedic knowledge, about prominent people, places, products etc. and basic relations of wide interest such as birthplaces, spouses, writing of books, acting in movies etc. Vertical domains have received some attention, including health, food, and consumer products. Yet another case are KBs about fictional works [Hertling and Paulheim, 2020, Labatut and Bost, 2019], such as Game of Thrones (GoT), the Marvel Comics (MC) universe, Greek Mythology or epic books such as War and Peace by Leo Tolstoy or the Cartel novels by Don Winslow. For KBs about fictional domains, the focus is less on basic relations like birthplaces or spouses, but more on relations that capture traits of characters and key elements of the narration. Relations of interest are allies, enemies, membership in clans, betrayed, killed etc.

Applications of fiction KBs foremost including supporting fans in entity-centric search. Some of the fictional domains have huge fan communities, and search engines frequently receive queries such as "Who killed Catelyn Stark?" (in GoT). Entity summarization is a related task, for example, a user asking for the most salient traits of Ygritte (in GoT). Although fiction serves to entertain, some of the more complex domains reflect sub-cultural trends and the zeitgeist of certain epochs. Analyzing their narrative structures and networks

of entities is of interest to humanities scholars. For example, superhero comics originated in the 1940s and boomed in post-war years, reflecting that era's zeitgeist (revived now). War and Peace has the backdrop of the Napoleonic wars in Russia, and the Cartel trilogy blends facts and fiction about drug trafficking. KBs enable deeper analyses of such complex texts for historians, social scientists, media psychologists and cultural-studies scholars.

**State of the Art and its Limitations:** RE with pre-specified relations for canonicalized entities is based on distant supervision via pre-compiled seed triples [Mintz et al., 2009, Suchanek et al., 2009]. Typically, these training seeds come from initial KBs, which in turn draw on Wikipedia infoboxes. The best RE methods are based on this paradigm of distant supervision, leveraging it for neural learning (e.g., [Soares et al., 2019, Wang et al., 2020, Yao et al., 2019, Zhang et al., 2017, Han et al., 2020]). They work well for basic relations, as there is no shortage of training samples (e.g., for birthplace or spouse). One of their key limitations is the bounded size of input text passages, typically a few hundred tokens only. This is not a bottleneck for basic relations where single sentences (or short paragraphs) with all three SPO components are frequent enough (e.g., in the full text of Wikipedia articles). However, for RE with non-standard relations over long fictional texts such as entire books, these limitations are major bottlenecks, if not showstoppers. This paper addresses the resulting challenges (also included among the open challenges in the overview by [Han et al., 2020]):

- How to go about distant supervision for RE targeting non-standard relations that have only few seed triples?

- How to cope with very long input texts, such as entire books, where relevant cues for RE are spread across passages?

**Approach and Contributions:** This paper presents a complete methodology and system for relation extraction from long fictional texts, called *KnowFi* (Knowledge extraction from Fictional texts). Our method leverages semi-structured content in wikis of fan communities on `fandom.com` (aka `wikia.com`). We extract an initial KB of background knowledge for 142 popular domains (TV series, movies, games). This serves to identify interesting relations and to collect distant supervision samples. Yet for many relations this results in very few seeds. To overcome this sparseness challenge and to generalize the training across the wide variety of relations, we devise a similarity-based ranking technique for matching seeds in text passages. Given a long input text, KnowFi judiciously selects a number of context passages containing seed pairs of entities. To infer if a certain relation holds between two entities, KnowFi's neural network is trained jointly for all relations as a multi-label classifier.

Extensive experiments with long books on five different fictional domains show that KnowFi clearly outperforms state-of-the-art RE methods. Even on conventional short-text benchmarks with standard relations, KnowFi is competitive with the best baselines. As an extrinsic use case, we demonstrate the value of KnowFi's KB for the task of entity summarization. The paper's novel contributions are:

- a system architecture for the new problem of relation extraction from long fictional texts, like entire novels and text contents by fan communities (Section 3);

- a method to overcome the challenge of sparse samples for distant supervision for non-standard relations (Section 4).

- a method to overcome the challenge of limited input size for neural learners, by judiciously selecting relevant contexts and aggregating results (Section 5);

- a comprehensive experimental evaluation with a novel benchmark for relation extraction from very long documents (Section 6), with code and data release upon publication.

## 2. Related Work

**Relation Extraction (RE):** Early work on RE from text sources has used rules and patterns, (e.g., [Agichtein and Gravano, 2000, Craven et al., 1998, Etzioni et al., 2004, Reiss et al., 2008]), with pattern learning based on the principle of relation-pattern duality [Brin, 1998]. Open IE [Banko et al., 2007, Mausam, 2016, Stanovsky et al., 2018] uses linguistic cues to jointly infer patterns and triples, but lacks proper normalization of SPO arguments. RE with pre-specified relations, on the other hand, is usually based on distant supervision via pre-compiled seed triples [Mintz et al., 2009, Suchanek et al., 2009]. A variety of methods have been developed on this paradigm, from probabilistic graphical models (e.g., [Pujara et al., 2015, Sa et al., 2017]) to deep neural networks (e.g., [Soares et al., 2019, Wang et al., 2020, Yao et al., 2019, Zhang et al., 2017, Han et al., 2020]). Distantly supervised neural learning has become the method of choice, with different granularities.

**Sentence-level RE:** Most neural methods operate on a per-sentence level. Distant-supervision samples of SPO triples serve to identify sentences that contain an entity pair (S and O) which stand in a certain relation. The sentence is then treated as a positive training sample for the neural learner. At test-time, the trained model can tag entity mentions and predict if the sentence expresses a given relation or not. This basic architecture has been advanced with bi-LSTMs, attention mechanisms and other techniques (e.g., [Zhang et al., 2017, Cui et al., 2018, Trisedya et al., 2019]). A widely used benchmark for sentence-level RE is TacRed [Zhang et al., 2017].

With recent advances on pre-trained language models like BERT [Devlin et al., 2019a] (or ElMo, GPT-3, T-5 and the like), the currently best RE methods leverage this asset for representation learning [Shi and Lin, 2019, Soares et al., 2019, Wadden et al., 2019, Yu et al., 2020].

**Document-level RE:** To expand the scope of inputs, Wang et al. [2019] proposed RE from documents, introducing the DocRed benchmark. However, the notion of documents is still very limited in size, given the restrictions in neural network inputs, typically around 10 sentences (e.g., excerpts from Wikipedia articles). Wang et al. [2020] is a state-of-the-art method for this document-level RE task, utilizing BERT and graph convolutions for representation learning. Zhou et al. [2021] further enhanced this approach. None of these methods can handle input documents that are larger than a few tens of sentences. KnowFi is the first method that is geared for book-length input.

**Fiction Knowledge Bases:** Understanding characters in literary texts and constructing networks of their relationships and interactions has become a small topic in NLP (e.g., [Chaturvedi et al., 2016, Labatut and Bost, 2019, Srivastava et al., 2016]). The work of [Chu et al., 2019, 2020] has advanced this theme for entity typing and type taxonomies for

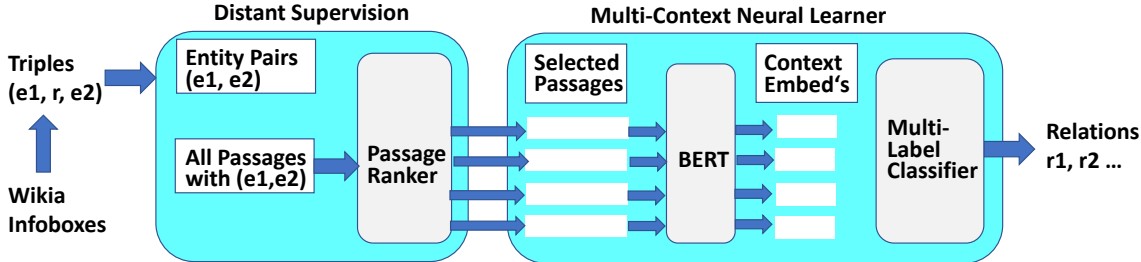

Figure 1: Overview of the KnowFi architecture.

fictional domains. However, this work does not address learning relations between entities for KB population.

The DBkWik project [Hertling and Paulheim, 2020] has leveraged structured infoboxes of fan communities at wikia (now renamed to `fandom.com`), to construct a large KB of fictional characters and their salient properties. However, this is strictly limited to relations and respective instances that are present in infoboxes. Our work leverages wikia infoboxes for distant supervision, but our method can extract more knowledge from a variety of text sources, including storylines and synopses by fans and, most demandingly, the full text of entire books.

## 3. System Overview

The architecture of the KnowFi system is illustrated in Figure 1. There are two major components:

- **Distant supervision** involves pre-processing infoboxes from Wikia-hosted fan communities, to obtain seed pairs of entities. These are used to retrieve relevant passages from the underlying text corpora: either synopses of storylines in Wikia or full-fledged content of original books. As the number of passages per entity pair can be very large in books, we devise a judicious ranking of passages and feed only the top-k passages into the next stage of training the neural network. Details are in Section 4.

- **Multi-context neural learning** feeds the top-k passages, with entity markup, jointly into a BERT-based encoder [Devlin et al., 2019b]. On top of this representation learning, a multi-label classifier predicts the relations that hold for the input entity pair. Details are in Section 5.

Note that a passage can vary from a single sentence to a long paragraph. The two seed entities would ideally occur in the same sentence, but there are many cases where they are one or two sentences apart. Figure 2 shows example texts from a GoT synopses in Wikia and from one of the original books.

The pre-processing of Wikia infoboxes resulted in 2.37M SPO triples for ca. 8,000 different relation names between a total of 461.4k entities, obtained from 142 domains (movie/TV series, games etc.). This forms our background knowledge for distant supervision. For obtaining matching passages, we focused on the 64 most frequent relations, including friend, ally, enemy and family relationships. Note that this stage is not domain-specific. Later we apply the learned model to specific domains such as GoT or Marvel Comics.

**Excerpt from Game of Thrones synopses at Wikia:**
*Eighteen years before the War of the Five Kings, Rhaegar Targaryen allegedly abducted Lyanna Stark in a scandal that led to the outbreak of Robert's Rebellion. Rhaegar eventually returned to fight n the war, but not before leaving Lysanna behind at the Tower of Joy, guarded by Lord Commander Gerold Hightower and Ser Arthur Dayne of the Kingsguard. Eddard Stark rode to war along her betrothed, Robert Baratheon, to rescue **his sister** and avenge the deaths of their father and brother at the orders of Aerys II, the Mad King.*

**Excerpt from Harry Potter book:**
*Harry had been a year old the night that Voldemort - the most powerful **Dark** wizard for a century, a wizard who had been gaining power steadily for eleven years, arrived at his house and **killed his father and mother**. Voldemort had then **turned his wand on** Harry; he had performed the **curse** that had **disposed** of many full-grown witches and wizards in his steady rise to power and, incredibly, it had not worked. Instead of **killing** the small boy, the curse had rebounded upon Voldemort. Harry had **survived** with nothing but a lightning-shaped cut on his forehead, and Voldemort had been **reduced** to something **barely alive**.*

Figure 2: Examples of input texts.

## 4. Distant Supervision with Passage Ranking

The KnowFi approach to distant supervision differs from prior works in two ways:

- **Passage ranking:** Identifying the best passages that contain seed triples, by judicious ranking, and using only the top-k passages as positive training samples.

- **Passages with gaps:** Including passages where the entities of a seed triple merely occur in separate sentences with other sentences in between.

**Passage ranking:** Seed pairs of entities are matched by many sentences or passages in the input corpora. For example, the pair (Herminone, Harry) appears in 1539 sentences in the the seven volumes of the Harry Potter series together. Many of these contain cues that they stand in the `friends` relation, but there are also many sentences where the co-occurrence is merely accidental. This is a standard dilemma in distant supervision for multi-instance learning [Riedel et al., 2010, Li et al., 2020]. Our approach is to identify the best passages among the numerous matches, by judicious ranking on a per-relation basis.

For each relation, we build a *prototype representation* by selecting sentences that contain lexical matches of all three SPO arguments, where the predicate is matched by its label in the background knowledge or a short list (average length of three) of synonyms and close hyponyms or hypernyms (e.g., "allegiance" or "loyalty" matching `ally`), manually collected from WordNet. Newly seen passages for entity pairs can then be scored against the per-relation prototypes by casting both into tf-idf-weighted bag-of-word models (or alternatively, word2vec-style embeddings) and computing their cosine distance. This way, we rank candidate passage for each seed pair and target relation.

**Passages with gaps:** Unlike encyclopedic articles, long texts on fictional domains have a narrative style where single sentences are unlikely to give the full information in the most compact way. Therefore, we consider multi-sentence contexts where entity mentions across different sentences. In addition to simple paragraphs, we consider passages with gaps where we include sentences that are not necessarily contiguous but leave out uninformative sentences. This way, we maximize the value of limited-size text spans fed into the neural

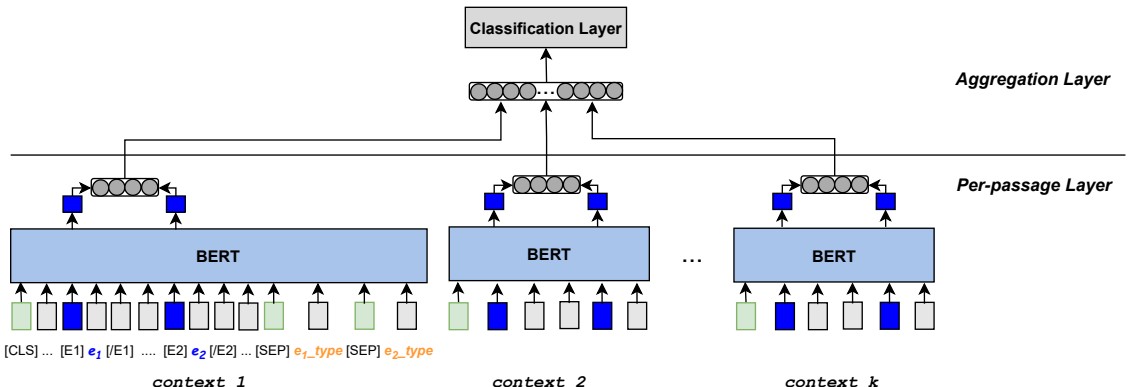

Figure 3: Neural network architecture for multi-context RE.

learner. This is in contrast to earlier techniques that consume whole paragraphs and rely on attention mechanism for giving higher weight to informative parts.

KnowFi has two configuration parameters: the maximum number of sentences allowed between sentences that contain seed entities, and the number of sentences directly preceding or following the occurrence of a seed entity. In our experiments, we include text where the two entities appear at most 2 sentences apart and 1 preceding and 1 following sentence for each of the entity mentions, up to 512 tokens which is the current limit of BERT-based networks.

**Negative training samples:** In addition to the positive training samples by the above procedure, we generate negative samples by the following random process. For each relation $r$, we pick random entities $e1$ and $e2$ for each of the S and O roles such that there are other entities $x$ and $y$ for which the background knowledge asserts (e1, r, x) and (y, r, e2) with $x \neq e2$ and $y \neq e1$. This improves on the standard technique of simply choosing any pair $e1, e2$ for which $(e1, r, e2)$ does not hold, by selecting more difficult cases and thus strengthening the learner. For example, both Herminone and Malfoy have some friends, but they are not friends of each other. The training of KnowFi uses a 1:1 ratio of positive to negative samples.

## 5. Multi-Context Neural Extraction

KnowFi is trained with and applicable to multiple passages as input to an end-to-end Transformer-based network with full backpropagation of cross-entropy loss. Our neural architecture has two specific components: a per-passage layer to learn BERT-based representations for each passage, and an aggregation layer that combines the signals from all input passages. In the experiments in this paper, the aggregation layer is configured to concatenate the representations of all passages, but other options are feasible, too.

Each input passage is encoded with markup of entity mentions. In addition, we determine semantic types for the entities, using the SpaCy tool (`https://spacy.io/`) that provides one type for each mention, chosen from a set of 18 coarse-grained types (person,

| Dataset | # Instances | # Rel. | # Pos. Inst. | # Neg. Inst. | avg. # Pos. Inst./Rel. | avg. # Pas./Inst. |
|---|---|---|---|---|---|---|
| Train | 81,025 | 64 | 40,920 | 40,105 | 640 | 1.5 |
| Dev | 20,257 | 64 | 10,363 | 9,894 | 162 | 1.5 |

Table 1: Statistics on training and validation set. (Rel.: relation, Inst.: instances, Pos.: positive instances, Neg.: negative instances, avg. #Pos.Inst./Rel.: average number of positive instance per relation, avg. #Pas./Inst.: average number of passages per instance)

| Universe | # rel. | # facts | top relations |
|---|---|---|---|
| Lord of the Rings | 13 | 1,143 | race, culture, realm, weapon |
| Game of Thrones | 18 | 2,547 | ally, culture, title, religion |
| Harry Potter | 20 | 4,706 | race, ally, house, owner |
| Dune | 11 | 133 | homeworld, ruler, commander |
| War and Peace | 10 | 101 | relative, child, spouse, sibling |

Table 2: Statistics on test data of the five test universes.

nationality/religion, event, etc.). The type of each entity mention in a passage is appended to the input vector. Figure 3 illustrates the neural network for multi-context RE.

## 6. LoFiDo Benchmark

To evaluate RE from long documents, we introduce the LoFiDo corpus (Long Fiction Documents). We compile SPO triples from infoboxes of 142 Wikia fan communities. After cleaning extractions and clustering synonyms, we obtain a total of 64 relations such as enemy, friend, ally, religion, weapon, ruler-of, etc.

For evaluating KnowFi and various baselines, we focus on 5 especially rich and diverse domains (i) Lord of the Rings (a series of three epic novels by J.R.R Tolkien), (ii) A Song of Ice and Fire (a series of five fantasy novels by George R.R. Martin, well-known for the Game of Thrones TV series based on it), (iii) Harry Potter (a series of seven books, written by J.K Rowling), (iv) Dune (a science-fiction novel by Frank Herbert), and (v) War and Peace (a classic novel by Leo Tolstoy). For the first four, Wikia infoboxes provide ground truth; for War and Peace, we manually crafted a small ground-truth KB. 20% of the triples from each of these universes are withheld for testing. For the first four domains, we consider both original novels as well as narrative synopses from Wikia as input sources. War and Peace is not covered by Wikia.

**LoFiDo Statistics.** Our LoFiDo corpus contains 81,025 instances for training and 20,257 instances for validation. For testing, we use five specific universes, which take input from both books and Wikia texts. The total number of instances in the test data from Wikia texts is 14,610, while in the case of books, it is 64,120. Ground-truth data for five test universes are provided for evaluation. Table 1 shows statistics on the training and validation data, while Table 2 shows statistics on the ground-truth of five domains in the test data. Further details on this dataset are in Appendix A and B. Code and data of KnowFi are available at https://www.mpi-inf.mpg.de/yago-naga/knowfi.

## 7. Experiments

### 7.1 Setup

**Baselines.** We compare KnowFi to three state-of-the-art baselines on RE:

- **BERT-Type** [Shi and Lin, 2019] which uses BERT-based encodings augmented with entity type information, also based on SpaCy output in our experiments for fair comparison.

- **BERT-EM** [Soares et al., 2019] which include entity markers in input sequences;

- **GLRE** [Wang et al., 2020] which additionally computes global entity representations and uses them to augment the text sequence encodings.

The first two baselines run on a per-sentence basis, whereas GLRE is a state-of-the art method for extractions from short documents, which we train on paragraph-level inputs. The inputs for these models (i.e. sentences or paragraphs) are randomly selected.

**KnowFi Parameters.** For passage selection, we rely on TF-IDF-based bag-of-words similarity between the passages and relation contexts, where each relation context contains the top-100 tokens, based on their tfidf scores. For selecting passages as multi-context input, we compute the cosine between tf-idf-based vectors of each passage against the relation-specific prototype vector; we select all passages with cosine above 0.5 as positive training samples. For the neural network, we use $\text{BERT}_{LARGE}$ (https://huggingface.co/transformers/model_doc/bert.html) with 24 layers, 1024 hidden size and 16 heads. The learning rate is $5e - 5$ with Adam, the batch size is 8, and the number of training epochs is 10.

**Evaluation Metrics.** The evaluation uses standard metrics like precision, recall and F1, averaged over all extracted triples. We report micro-averaged numbers for all relations together, and drill down on selected relations of interest. In addition, we report numbers for HITS@k and MRR. As ground-truth, we perform two different modes of evaluation:

- **Automated evaluation** is based on ground-truth from Wikia infoboxes. This is demanding on precision, but penalizes recall because of its limited coverage.

- **Manual evaluation** is based on obtaining assessments of extracted triples via crowdsourcing. This way, we include correct triples that are not in Wikia infoboxes, and thus achieve higher recall.

### 7.2 Results

**Automated Evaluation.** Table 3 shows average precision, recall and F1 score. We can see that sentence-level baselines achieve comparatively high coverage, due to considering every sentence. Yet their precision is extremely low. GLRE and KnowFi achieve much higher precision, though GLRE fails to achieve competitive recall, presumably because its training on all paragraphs lowers its predictive power. As an illustration, GLRE produces only 173 assertions from all Harry Potter books, while KnowFi produces 600.

We also observe that for all methods, extraction from books is considerably harder than from the more concise synopses in Wikia.

In addition to the P/R/F1 scores, in Table 4 we also take an entity-centric view and evaluate how well correct extractions rank. The HITs@k metric reports how often a correct result appears among the top extractions per entity-relation pair (e.g., among top-5

| Models | Books | | | Wikia Texts | | |
|---|---|---|---|---|---|---|
| | Precision | Recall | F1-score | Precision | Recall | F1-score |
| **BERT-Type** (Shi and Lin) | 0.00 | 0.07 | 0.00 | 0.02 | 0.05 | 0.00 |
| **BERT-EM** (Soares et al.) | 0.06 | 0.11 | 0.08 | 0.11 | 0.20 | 0.14 |
| **GLRE** (Wang et al.) | **0.17** | 0.03 | 0.05 | **0.18** | 0.07 | 0.10 |
| **KnowFi** | 0.14 | **0.11** | **0.12** | 0.17 | **0.26** | **0.21** |

Table 3: Automated evaluation: average precision, recall and F1 scores.

| Models | Books | | | | Wikia Texts | | | |
|---|---|---|---|---|---|---|---|---|
| | HIT@1 | HIT@3 | HIT@5 | MRR | HIT@1 | HIT@3 | HIT@5 | MRR |
| **BERT-Type** (Shi and Lin) | 0.01 | 0.02 | 0.04 | 0.02 | 0.09 | 0.20 | 0.23 | 0.16 |
| **BERT-EM** (Soares et al.) | 0.24 | 0.35 | 0.37 | 0.35 | 0.49 | 0.59 | 0.61 | 0.54 |
| **GLRE** (Wang et al.) | 0.40 | 0.53 | 0.54 | 0.46 | 0.47 | 0.62 | 0.68 | 0.57 |
| **KnowFi** | **0.45** | **0.54** | **0.55** | **0.50** | **0.60** | **0.71** | **0.72** | **0.66** |

Table 4: Automated evaluation: average *HIT@K* and *MRR* scores.

extracted enemies of Harry Potter), while MRR reports the mean reciprocal rank of the first extraction. We can observe that KnowFi outperforms all baselines on both metrics.

**Manual Evaluation.** The low absolute scores in the above evaluation largely stem from incomplete automated ground truth. We therefore conducted an additional manual evaluation. For each domain, we select top 100 extractions from the results and used crowdsourcing to manually label their correctness. The annotators were Amazon master workers with all time approval rate > 90%, and additional test questions were used to filter responses. We observed high inter-annotator agreement, on average of 0.81.

Table 5 shows results of our manual evaluation on four domains (Dune was left out due to complexity). As one can see, KnowFi outperforms the baselines on most input texts, and achieves a remarkable precision on both books and wikia texts (average of 0.57 on books and 0.73 on wikia texts).

We repeat the entity-centric evaluation with manual labels for three relations of special interest in fiction, *friend*, *enemy* and *ally*. We select 10 popular entities each from LoTR, GoT and Harry Potter. The resulting precision scores are shown in Table 6. As one can see, KnowFi is achieves high precision among its top extractions, e.g., 78% and 73% precision at rank 1 for *friend* assertions from books/Wikia texts.

**Evaluation on Short-Text Datasets.** To evaluate the robustness of KnowFi, we also evaluate its performance on the existing sentence-level RE dataset TACRED, and the short document-level RE dataset DocRED. The results are shown in Tbl. 7. We find KnowFi's

| Models | Books | | | | | Wikia Texts | | | | |
|---|---|---|---|---|---|---|---|---|---|---|
| | LoTR | GOT | HP | WP | **Avg.** | LoTR | GOT | HP | WP | **Avg.** |
| **BERT-Type** (Shi and Lin) | 0.01 | 0.54 | 0.09 | 0.11 | 0.19 | 0.09 | 0.12 | 0.15 | 0.19 | 0.14 |
| **BERT-EM** (Soares et al.) | 0.45 | 0.66 | 0.37 | 0.29 | 0.44 | 0.70 | 0.78 | 0.48 | 0.50 | 0.62 |
| **GLRE** (Wang et al.) | 0.27 | 0.25 | **0.56** | 0.47 | 0.39 | 0.37 | 0.56 | **0.71** | 0.56 | 0.55 |
| **KnowFi** | **0.45** | **0.76** | 0.55 | **0.50** | **0.57** | **0.71** | **0.83** | 0.71 | **0.67** | **0.73** |

Table 5: Manual evaluation - average precision scores over 4 input texts (*LoTR:* Lord of the Rings, *GOT:* Game of Thrones, *HP:* Harry Potter, *WP:* War and Peace).

| Sources | friend (top $k$ objects) | | | enemy (top $k$ objects) | | | ally (top $k$ objects) | | |
|---|---|---|---|---|---|---|---|---|---|
| | **1** | **3** | **5** | **1** | **3** | **5** | **1** | **3** | **5** |
| **Books** | 0.78 | **0.82** | 0.80 | **0.55** | 0.45 | 0.47 | 0.63 | **0.67** | 0.63 |
| **Wikia Texts** | 0.73 | **0.76** | 0.75 | **0.60** | 0.48 | 0.49 | **0.70** | 0.67 | 0.62 |

Table 6: Manual evaluation - precision of *friend*, *enemy* and *ally* relations.

| Models | TACRED | | DocRED | |
|---|---|---|---|---|
| | **F1 - Dev** | **F1 - Test** | **F1 - Dev** | **F1 - Test** |
| **BERT-Type** (Shi and Lin) | 0.65 | 0.64 | - | - |
| **BERT-EM** (Soares et al.) | 0.64 | 0.62 | - | - |
| **GLRE** (Wang et al.) | - | - | - | **0.57** |
| **KnowFi** | **0.67** | **0.66** | 0.52 | 0.51 |

Table 7: Automated evaluation - short text datasets TACRED and DocRED.

performance on TACRED is on par with BERT-Type and BERT-EM (0.66 test-F1, versus 0.63 and 0.62 for the baselines), the modest gain indicating that the combination of entity types and markers is beneficial. On DocRED, KnowFi achieved 0.51 F1-score, slightly below the GLRE model at 0.57 F1-score. We hypothesize that the modest losses stem from the fact that GLRE is specifically tailored for the short documents of TACRED, where multi-context aggregation is not relevant. At the same time, the single contexts GLRE considers have no inherent size limitation, unlike the 2-sentence distance threshold used in KnowFi.

**Ablation Study.** To evaluate the impact of passage ranking, we ran KnowFi *without passage ranking* for both training and prediction. Instead, passages were randomly selected. In automated evaluation, without passage ranking, KnowFi achieves comparable recall but lower precision: 0.07 vs. 0.14 on books and 0.12 vs. 0.17 on Wikia texts. This pattern is also observed in manual evaluation, where KnowFi, without passage ranking, achieves a precision of 0.43 vs. 0.57 on books and 0.55 vs. 0.73 on Wikia texts. Further experiments and ablation studies are in Appendix C and D.

**Error Analysis.** The precision gain from automated to manual evaluation (Table 3 vs. Table 5) indicates that ground-truth incompleteness is a confounding factor. By inspecting a sample of 50 false positives, we found that 20% originated from incomplete ground truth, while 54% were indeed not inferrable from the given contexts, pointing to limitations of the distant-supervision-based training (e.g., extracting friendship from the sentence *"Thorin came to Bilbo's door"*). Another 15% were errors in determining the subject or object in complex sentences with many entity mentions, e.g., extracting *friend(Boromir, Gimli)* from the sentence *"Boromir went ahead, Legolas and Gimli, who by now had become friends, followed."* Finally, 7% of the false positives captured semantically related relations but missed the correct ones.

By sampling false negatives, we found that in 52% of the cases the retrieved contexts did not allow the proper inference, indicating limitations in the context retrieval and ranking. In 33% of the cases, a human reader could spot the relation in the top-ranked contexts (e.g., `hasCulture (Legolas, Elf)` in *"He saw Legolas seated with three other Elves"*). Appendix E shows some anecdotal examples for the output of KnowFi and baselines.

| In **Lord of the Rings**, which summary is more informative for `Frodo Baggins`: | |
|---|---|
| **Summary 1:** | <Frodo, has parent, Drogo>, <Frodo, has culture, Shire>, <Frodo, has enemy, Sauron>, <Frodo, has friend, Sam>, <Frodo, has weapon, Sting> |
| **Summary 2:** | <Frodo, has owner, Gandalf>, <Frodo, has weapon, Ring>, <Frodo, has parent, Drogo>, <Frodo, has affiliation, Sam>, <Frodo, has culture, Marish> |

Table 8: Sample task for assessing entity summaries.

## 8. Extrinsic Use Case: Entity Summarization

To assess the salience in the extractions produced by KnowFi, we pursued a user study to compare entity summaries, one from KnowFi and one by randomly drawing from one of the three baselines. Each entity summary includes at most 5 best extractions (distinct relations) from the book series Lord of the Rings, Game of Thrones and Harry Potter. For each domain, we generate summaries for 5 popular entities. We give pairs of summaries, with randomized order, to Amazon master workers for selecting the more informative one. Table 8 shows an example of this crowdsourcing task. The annotators preferred KnowFi-based summaries over BERT-Type, BERT-EM and GLRE in 93%, 64% and 81% of the cases, respectively.

## 9. Conclusion

To the best of our knowledge, this paper is the first attempt at relation extraction (RE) from long fictional texts, such as entire books. The presented method, KnowFi, is specifically geared for this task by its judicious selection and ranking of passages. KnowFi outperforms strong baselines on RE by a substantial margin, and it performs competitively even on the short-text benchmarks TacRed and DocRed. The absolute numbers for precision and recall show that there is still a lot of room for improvement. This underlines our hypothesis that long fictional texts are a great challenge for RE. Our LoFiDo corpus of Wikia texts, book contents, and ground-truth labels will be made available to foster further research. All information can be found at https://www.mpi-inf.mpg.de/yago-naga/knowfi.

### Acknowledgments

We thank Matteo Cannaviccio and Filipe Mesquita from Diffbot Inc. for their advice and support on labeling data, and Diffbot Inc. for free access to their RE API. We also thank our anonymous reviewers for their valuable comments.

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

## Appendix A. Training Data Extraction

Many current KBs like Yago, DBpedia or Freebase have been built by extracting the information from infoboxes, category network and leveraging the markup language of Wikipedia. The relations of these KBs are then used as schema for many later supervised relation extractors. However, for fiction, Wikipedia has too low coverage of entities and relevant relations.

**Wikia.** Wikia (or Fandom) is the largest web platform for fiction and fantasy. It contains over 385k communities with total of over 50M pages. Each community (usually discusses about one fictional universe) is organized as a single Wiki. With a wide range of coverage on fiction and fantasy, Wikia is one of the 10 most visited websites in the US (as of 2020)[1].

**Crawling.** We download all universes which contain over 1000 content pages and have available dump files from Wikia, and get total of 142 universes in the end. From these universes, we extract all information from their category networks and infoboxes, and build a background knowledge base for each universe.

**Definition A.1.** *Background KB of an universe is a collection of entities, entity mentions, simple facts that describe relations between entities and a type system of the universe.*

**Background knowledge extraction.** To extract the background KBs, we follow a simple procedure:

- **Type system construction:** The type system is extracted from Wikia category network. We adapt the technique from the TiFi system [Chu et al., 2019] to structure and clean the type system.

- **Entity detection:** Entities and entity mentions can be easily extracted from the dump file. We consider page titles as the entities in the universe (except administration and category pages). On the other hand, entity mentions only appear in texts. By using Wiki markup, each mention can be extracted and linked to the entity with a confident score which is computed based on its frequency.

- **Infobox extraction:** Facts about each entity are extracted from its infobox. Infobox is presented in table format with the entity's attributes and their values. Each extracted fact is presented in a triple with subject, predicate (relations) and object. In particular, we consider the main entity as subject, the attributes as predicates, and the values as objects. We manually check if there is any misspelling in the relations and merge them if necessary.

This results an average of 158k entities and 13.5k facts in each universe. The information from these background KBs is then used for all three later steps.

**Relation Filtering.** After extracting the background KBs, we get all relations from the facts of all universes and consider them as relation candidates that can be extracted in fictional domains. However, beside meta relations which are not really related to the content of universes, such as *season, page, episode,...*, there is much noise in the relations since they are manually created by fans. To remove noise and keep popular relations, we do **relation filtering** as follows:

---

1. https://ahrefs.com/blog/most-visited-websites/

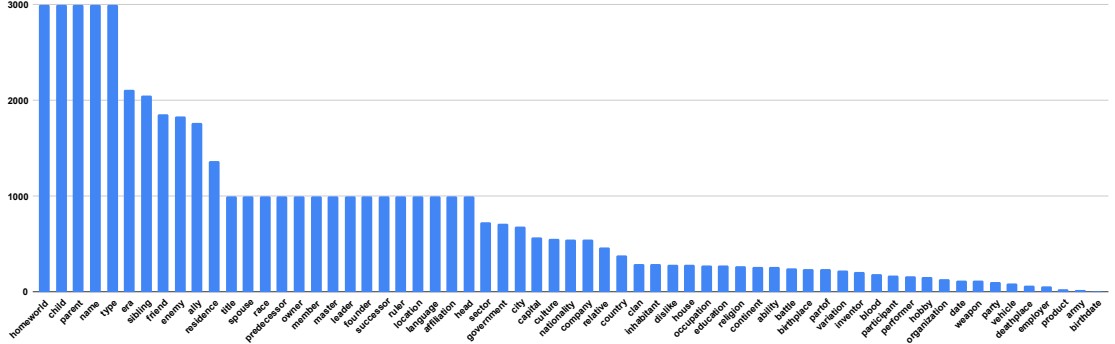

Figure 4: Statistics on training data.

- Pre-processing: a combination of stemming and keeping relations with length at least 3 (except for some relations like *job, age, son, etc.*).

- Infrequent-relation removing: we only keep relations which are in at least 5 universes and appear in over 20 facts.

- Meta-relation removing: we manually check if the relation is a meta-relation. In total, there are 247 relations considered as meta-relations.

- Misspelling detection: Misspelling relations are manually detected and grouped with the correct relations, for example, *affilation* and *affiliation*.

- Grouping: Synonym relations are manually grouped together, for example, *leader* and *commander*.

After relation filtering, we reduce the number of relations from over 8,000 to 64 relations. These relations are considered as *popular relations* in fictional domains and used as targets for the relation extraction step. We realize that in fictional domains, the relations expressing the friendly or hostile relationship between two entities are interesting, hence, we keep *friend* and *enemy* as two relations which are always extracted. Figure 4 shows statistics on training data. We publish the training data as supplementary material.

## Appendix B. Background KB Statistics

One of our contribution is the background KB dataset on popular universes in fictional domains. To have an overview about the dataset, Table 9 shows some statistics on our background KBs database, which include information about universes, entities, type systems, relations and facts.

From the 5 domains used for testing, the number of relations varies from 10 to 20, and the number of ground-truth triples varies from 1,100 to 4,700 for the first three domains, and was between 100 and 200 the last two.

| Statistics | | Top Universes | | Top Relations | | |
|---|---|---|---|---|---|---|
| # Universes | 142 | **Universes** | **# Facts** | **Relations** | **#** | **# universes** |
| | per universe | Star Wars | 282,440 | name | 238,290 | 111 |
| # Facts | 13,539 | Monster Hunter | 153,178 | type | 112,347 | 94 |
| # Relations | 163 | World of Warcraft | 144,586 | gender | 95,972 | 77 |
| # Entities | 158,066 | Marvel | 77,826 | affiliation | 85,676 | 61 |
| # Entity Mentions | 224,782 | DC Comics | 69,190 | era | 53,871 | 12 |
| # Entity Types | 1246 | Forgotten Realms | 63,360 | hair(color) | 50,325 | 41 |

Table 9: Statistics on background KBs.

| Threshold | Books | | | Wikia Texts | | |
|---|---|---|---|---|---|---|
| | **Precision** | **Recall** | **F1-score** | **Precision** | **Recall** | **F1-score** |
| **0.4** | 0.07 | **0.11** | 0.09 | 0.12 | **0.32** | 0.17 |
| **0.5** | 0.14 | **0.11** | 0.12 | 0.17 | 0.26 | **0.21** |
| **0.6** | **0.20** | 0.10 | **0.13** | **0.21** | 0.17 | 0.19 |

Table 10: Automated Evaluation: Study on the similarity threshold.

## Appendix C. Additional Experiments

**Similarity Threshold.** In our experiments, we consider all passages with cosine above 0.5 as positive training samples (section 7.1). To assess the effect of the similarity threshold, we conduct an ablation study on it. Table 10 reports the automated results of KnowFi on both books and Wikia texts, where the threshold varies. For the author response we completed two other runs (threshold 0.4 and 0.6) that indicate modest influence, for the final version we would provide insights for all threshold values from 0 to 1 (in 0.1 step size). The results show that, with a similarity threshold around 0.5, the model achieves the best F1. By increasing the similarity threshold, the model is able to achieve higher precision but lower recall and vice versa.

**Embedding-based Passage Ranking.** KnowFi uses a simple TF-IDF-based schema for passage ranking. To assess the effectiveness of this method, we conduct an ablation study on the ranking step. Instead of using TF-IDF, we compute the embeddings of the passages and the relation contexts using Sentence-BERT [Reimers and Gurevych, 2019]. The cosine similarity between the passage embedding and the relation context embedding is then computed using the *sklearn* library. We select all passages with cosine above 0.0 (range [-1,1]), as positive training samples, with maximum of 5 passages per each training instance. Table 11 shows the automated results of KnowFi on both books and Wikia texts. The higher scores on recall shows that the embeddings can help the model capture the semantic relationships between passages and relation contexts better, especially when handling the cases of synonymy, while TF-IDF only handles the cases of lexical matching. However, in general, both techniques are on par, and embeddings do not improve the results, in terms of F1-score.

**GLRE with Passage Ranking.** In our experimental setup, the inputs of GLRE [Wang et al., 2020] (for both train and test) are randomly selected. To assess the effect of passage ranking on GLRE, we conduct a study where the inputs of GLRE are selected by using our method for passage ranking. Table 12 shows that, by using passage ranking to filter the

| Ranking Methods | Books | | | Wikia Texts | | |
|---|---|---|---|---|---|---|
| | **Precision** | **Recall** | **F1-score** | **Precision** | **Recall** | **F1-score** |
| **TF-IDF-based** | **0.14** | **0.11** | **0.12** | **0.17** | 0.26 | **0.21** |
| **Embedding-based** | 0.12 | **0.11** | **0.12** | 0.10 | **0.30** | 0.15 |

Table 11: Automated Evaluation: Study on the ranking method.

| Ranking Methods | Books | | | Wikia Texts | | |
|---|---|---|---|---|---|---|
| | **Precision** | **Recall** | **F1-score** | **Precision** | **Recall** | **F1-score** |
| **GLRE** (Wang et al.) | 0.17 | 0.03 | 0.05 | 0.18 | 0.07 | 0.10 |
| **GLRE + Passage Ranking** | **0.20** | 0.03 | 0.06 | **0.21** | 0.10 | 0.13 |
| **KnowFi** | 0.14 | **0.11** | **0.12** | 0.17 | **0.26** | **0.21** |

Table 12: Automated Evaluation: GLRE with Passage Ranking.

inputs, GLRE is able to achieve higher precision and recall, compared to GLRE without passage ranking. However, this enhanced variant is still inferior to KnowFi by a substantial margin.

## Appendix D. Impact of Training Data Quality

Training data is one of the most important factors that impact the quality of the supervised models, therefore, it is essential to maintain the quality of the training data, especially when working on specific domains where the training data is usually not available. To evaluate the quality of our training data, we compare KnowFi and a variant (i.e., without using passage ranking on training data collection) with two other methods which are trained using manual training datasets:

- TACRED [Zhang et al., 2017], a popular dataset for relation extraction on the sentence level. We train our relation extraction model using TACRED and use the model to extract the relations from the test data.

- Diffbot [Mesquita et al., 2019], a commercial API for relation extractions. We run the Diffbot API on our test data to extract the relations.

Note that these comparisons are not systematic, as there are confounding other differences between these two methods and KnowFi. In particular, the comparison with TACRED also comes with a reduction to sentence-level training examples only, while the Diffbot API is a pre-trained general-purpose extractor with necessary limitations in more specific use cases. We automatically evaluate the extractions on three popular relations, *spouse, sibling, child*, since they are contained in all datasets.

**Results.** Table 13 reports the results on two universes, *Lord of the Rings* and *Game of Thrones*. The results show that, our training data achieves comparable results with other datasets and even higher F1-scores, in both books and Wikia texts.

| Universes | Models | Books | | | Wikia Texts | | |
|---|---|---|---|---|---|---|---|
| | | Precision | Recall | F1-score | Precision | Recall | F1-score |
| **LoTR** | **Diffbot** | 0.68 | 1.75 | 0.98 | 1.69 | 54.39 | 3.28 |
| | **TACRED-based** | **28.57** | 0.93 | 1.79 | 5.34 | 37.96 | 9.36 |
| | **KnowFi - w/o ranking** | 1.34 | **4.38** | 2.05 | 2.20 | **79.82** | 4.27 |
| | **KnowFi** | 15.1 | 2.00 | **3.53** | **8.19** | 27.19 | **12.58** |
| **GOT** | **Diffbot** | 6.10 | **18.97** | 9.24 | 7.85 | **61.46** | 13.92 |
| | **TACRED-based** | 8.45 | 4.61 | 5.96 | 19.66 | 40.11 | 26.39 |
| | **KnowFi - w/o ranking** | 8.29 | 15.81 | 10.87 | 9.64 | 47.43 | 16.03 |
| | **KnowFi** | **11.63** | 18.83 | **12.64** | **19.8** | 50.59 | **28.47** |

Table 13: Average scores on three popular relations: *spouse, sibling, child*

| Source | Relation | Context(s) | BERT-EM | GLRE | KnowFi | GT |
|---|---|---|---|---|---|---|
| **Books** | *enemy* | C1: So to gain time Gollum challenged Bilbo to the Riddle-game, saying that if he asked a riddle which Bilbo could not guess, then he would kill him and eat him. C2: There Gollum crouched at bay, smelling and listening; and Bilbo was tempted to slay him with his sword. | ✓ | ✗ | ✓ | - |
| | *weapon* | C1: They watched him rejoin the rest of the Slytherin team, who put their heads together, no doubt asking Malfoy whether Harry's broom really was a Firebolt. C2: Faking a look of sudden concentration, Harry pulled his Firebolt around and sped off toward the Slytherin end. C3: Harry was prepared to bet everything he owned, including his Firebolt, that it wasn't good news... | ✓ | ✗ | ✓ | - |
| | *ally* | C1:...Lord Blackwood shall be required to confess his treason and abjure his allegiance to the Starks ... C2:..."I swore an oath to Lady Stark, never again to take up arms against the Starks", said Blackwood ... | ✗ | ✗ | ✓ | ✓ |
| | *founder* | There was a great roar and a surge toward the foot of the stairs; he was pressed back against the wall as they ran past him, the mingled members of the Order of the Phoenix, Dumbledore's Army, and Harry's old Quidditch team, all with their wands drawn, heading up into the main castle. | ✗ | ✓ | ✓ | ✓ |
| **Wikia Texts** | *friend* | Mulciber was also a friend of Severus Snape, which upset Lily Evans, who was Snape's best friend at the time. | ✓ | ✗ | ✓ | - |
| | *spouse* | ...Later, after sweets and nuts and cheese had been served and cleared away, Margaery and Tommen began the dancing, looking more than a bit ridiculous as they whirled about the floor. The Tyrell girl stood a good foot and a half taller than her little husband, and Tommen was a clumsy dancer at best ... | ✗ | ✓ | ✓ | ✓ |
| | *weapon* | Randyll repeatedly berates Sam: he insults his weight, tells him the Night's Watch failed to make a man out of him, and says he will never be a great warrior , or inherit Heartsbane, the Tarly family's ancestral Valyrian steel sword. | ✓ | ✗ | ✓ | ✓ |
| | *culture* | C1: The most powerful Ainu, Melkor (later called Morgoth or "Dark Enemy" by the elves), Tolkien's equivalent of, disrupted the theme, and in response, Eru Ilúvatar introduced new themes that enhanced the music beyond the comprehension of the Ainur. C2: Melkor's brother was Manwë, although Melkor was greater in power and knowledge than any of the Ainur. | ✓ | ✓ | ✓ | ✓ |

Table 14: Anecdotal examples for the outputs of KnowFi (**GT**: ground-truth, subject in red, object in blue).

# Appendix E. Anecdotal Examples

Table 14 gives examples for the output of the various methods on sample contexts. The red color texts denote subjects and the blue color texts denote objects.

