# OpenReview forum: "KnowFi: Knowledge Extraction from Long Fictional Texts"
_AKBC.ws/2021/Conference — AKBC 2021_

### Official Review · Reviewer_2wDt · 2021-07-20
**A great step towards KB construction from long docs.**

**Rating:** 7
**Confidence:** 4

**Review:**

The paper focuses on the relation extraction task from long fictional texts (i.e., book-level RE), which is an important step towards KB construction from books. The authors present a neural architecture to address the challenges such as sparsity, distant-supervision, size limitation, etc. Although the proposed method is not particularly novel, the paper presents a nice first work in this direction. The authors also provide a new benchmark LoFiDo for evaluating different methods, which is also a great contribution.

I don't have any major concerns about this paper.

---

> ### Author Response · Authors · 2021-07-30
> **Thank you very much!**
>
> Thank you for your comments on our work, we appreciate your time and effort!

---

### Official Review · Reviewer_NafS · 2021-07-22
**A new task and benchmark for fictional relation extraction, but missing some details**

**Rating:** 6
**Confidence:** 4

**Review:**

Strengths:
1. A new task and benchmark: This paper focuses on a new task to extract relations from long fictional texts, and release a new benchmark with KB annotations from Wikia or manual annotations.
2. Model design is reasonable: To solve the sparse issue of distant supervision, the authors rank passages based on TF-IDF; to tackle the long text input issue, multiple passages are encoded and are then concatenated.
3. The experiments show the superiority of the proposed model.

Weakness and detailed comments:
1. Missing model and experiment details.
    - 1.1 Lack of details of "prototype representation". How long is the "short list of synonyms and close hyponyms or hypernyms"? Where did you get the "hyponyms or hypernyms"?
    - 1.2 According to "the aggregation layer is configured to concatenate the representations of all passages", the number of passages should be fixed. Otherwise, it cannot match the dimension of the following classification layer. However, in the experiment, it is said that "we select all passages with cosine above 0.5 as positive training samples", which seems that the number of passages is not fixed, but is decided by the similarity threshold.
    - 1.3 What do your mean by "context selection" in the parameter setting. There is no such terminology in the model sections. Do you mean the "Passage ranking"?
    - 1.4 Also, I cannot follow "chosing the top-100 tokens per relation as its context".
2. Missing dataset details.
    - 2.1 There are no data statistics in both the main paper and the Appendix, which is important for a newly released dataset.
    - 2.2 Will the passages that you selected be released?
    - 2.3 About "20% of the triples from each of these universes are withheld for testing.", how about the validation set? It also does not mention whether parameters are selected on the validation set.
3. Questions about the experiments.
    - 3.1 Why do you choose TF-IDF to rank passages? Did you compare with embedding-based ranking method? It seems that the ranking step is the most crucial step in your model, but TF-IDF model cannot handle synonyms.
    - 3.2 In section 8, "we pursued a user study to compare entity summaries, one by KnowFi and one by a baseline", which conflicts with "We compare KnowFi to all baselines." Does the first sentence is written by mistake?
4. Additional questions: To deal with long text using Transformer-based encoders, the models designed for long text such as Longformer should be a natural thought. Have you tried that?

Typos:
- "e1,e2 for which (e1,r,e2 does not hold" --> "e1,e2 for which (e1,r,e2) does not hold"

---

> ### Author Response · Authors · 2021-07-30
> **Thank you for your valuable feedback!**
>
> **Missing model and experiment details.**
> > 1.1 Lack of details of "prototype representation". How long is the "short list of synonyms and close hyponyms or hypernyms"? Where did you get the "hyponyms or hypernyms"?
>
> Answer: We manually select the synonyms/hypernyms/hyponyms for each relation, based on WordNet and the relation filtering step (e.g. relation grouping - Appendix A). On average, each list contains about 3 synonyms/hypernyms/hyponyms.
>
> > 1.2 According to "the aggregation layer is configured to concatenate the representations of all passages", the number of passages should be fixed. Otherwise, it cannot match the dimension of the following classification layer. However, in the experiment, it is said that "we select all passages with cosine above 0.5 as positive training samples", which seems that the number of passages is not fixed, but is decided by the similarity threshold.
>
> Answer: The passages are first pre-filtered by the similarity threshold. Then, if more than k passages survive this filter, we pick only top-k passages as input of the neural model (k = 5 in our experiment)
>
> > 1.3 What do you mean by "context selection" in the parameter setting. There is no such terminology in the model sections. Do you mean the "Passage ranking"?
>
> Answer: Yes, it should be "passage ranking" or "passage selection". We revised the terminology in the paper.
>
> > 1.4 Also, I cannot follow "choosing the top-100 tokens per relation as its context".
>
> Answer: For each relation, we select top-100 tokens (based on their TF-IDF scores) as the context. We clarified this in the paper.
>
> **Missing dataset details.**
> > 2.1 There are no data statistics in both the main paper and the Appendix, which is important for a newly released dataset.
>
> Answer: The dataset includes 81,025 instances for training and 20,257 for validation, extracted from Wikia. For the test data, we use five specific universes, which take input from both books and Wikia texts. The total number of instances in the test data from Wikia texts is 14,610 and from books is 64,120. Thanks for this comment, we have added the statistics on LoFiDo in appendix B (LoFiDo Statistics) in the revised paper.
>
>
> > 2.2 Will the passages that you selected be released?
>
> Answer: The passages will be released with the final paper, along with the code.
>
> > 2.3 About "20% of the triples from each of these universes are withheld for testing.", how about the validation set? It also does not mention whether parameters are selected on the validation set.
>
> Answer: Please refer to the answer of question 2.1.
>
> **Questions about the experiments.**
> > 3.1 Why do you choose TF-IDF to rank passages? Did you compare with embedding-based ranking method? It seems that the ranking step is the most crucial step in your model, but TF-IDF model cannot handle synonyms.
>
> Answer: Based on your suggestion, we also conducted another study on the ranking step, where we use contextualized embeddings. The details about this study have been added in appendix D (Additional Experiments). As one can see, using embedding-based ranking, the model can somewhat capture the case of synonyms in the context, hence achieves higher recall. However, in general, both techniques are on par, and embeddings do not improve the results, in terms of F1-score.
>
> > 3.2 In section 8, "we pursued a user study to compare entity summaries, one by KnowFi and one by a baseline", which conflicts with "We compare KnowFi to all baselines." Does the first sentence is written by mistake?
>
> Answer: Each summary pair is generated by picking one from KnowFi and one by randomly drawing from one of the three baselines. We clarified this in the paper.
>
> > Additional questions: To deal with long text using Transformer-based encoders, the models designed for long text such as Longformer should be a natural thought. Have you tried that?
>
> Answer: Thanks for the suggestion. This is left for future work.

---

### Official Review · Reviewer_UK5j · 2021-07-22
**A new and interesting benchmark, but more details should be included**

**Rating:** 6
**Confidence:** 4

**Review:**

Summary: This paper proposed a novel benchmark for relation extraction from very long documents and a new method -- KnowFi to deal with the problem. The dataset is constructed from Wikia containing long fictional texts. The method, KnowFi, generally contains two modules -- passage ranking and multi-context neural extraction. They first rank the source passages via a similarity function and then feed the top-$k$ passages to a BERT-based multi-context neural extraction module to get its relation. Experiments showed that KnowFi achieved strong performance on this new benchmark and two conventional short-text benchmarks.

Strengths:
1. The benchmark proposed in this work is novel and worth exploring.
2. The proposed method -- KnowFi is simple yet effective.
3. Strong results on this benchmark and two conventional short-text benchmarks.

Weaknesses:
1. The statistics of the constructed dataset -- LoFiDo, are missing. How many samples do you use for training and evaluation? How many passages are given for each sample?
2. Lacking ablations of the Multi-Context Neural Extraction module, how does the similarity threshold which is set to 0.5, affect the performance?

Questions:
1. Have you tried any other baselines for passage ranking? For example, matching SPO by word embeddings or contextualized word embeddings.
2. What do you mean by "For context selection, we rely on TF-IDF-based bag-of-words similarity, choosing the top-100 tokens per relation as its context."?
3. Have you tried the comparison of the proposed method with GLRE + the passage ranking?

---

> ### Author Response · Authors · 2021-07-30
> **Thank you for your valuable feedback!**
>
>
> > The statistics of the constructed dataset -- LoFiDo, are missing. How many samples do you use for training and evaluation? How many passages are given for each sample?
>
> Answer: The dataset includes 81,025 instances for training and 20,257 for validation. On average, each sample contains ~1.5 passages. For the test data, we only test on five specific universes, which take the input from both books and Wikia texts. The total number of instances in the test data from Wikia texts is 14,610 and from books is 64,120. Thanks for this comment, we have added the statistics on LoFiDo in appendix B (LoFiDo Statistics) in the revised paper.
>
>
> > Lacking ablations of the Multi-Context Neural Extraction module, how does the similarity threshold which is set to 0.5, affect the performance?
>
> Answer: We pick 0.5 as the default threshold (possible values from 0 to 1). Based on your suggestion, we conducted an ablation study on the similarity threshold. We have added the results of this study in appendix D (Additional Experiments) in the revised paper. The results show that, with a similarity threshold around 0.5, the model achieves the best F1. By increasing the similarity threshold, the model can achieve higher precision but lower recall and vice versa.
>
>
>
> > Have you tried any other baselines for passage ranking? For example, matching SPO by word embeddings or contextualized word embeddings.
>
> Answer: Thanks for this suggestion. We have now conducted another study on the ranking step, where we use contextualized embeddings. The details about this study have been added in appendix D (Additional Experiments). As one can see, using embedding-based ranking, the model can somewhat capture the case of synonyms in the context, hence achieves higher recall. However, in general, both techniques are on par, and embeddings do not improve the results in terms of F1-score.
>
> > What do you mean by "For context selection, we rely on TF-IDF-based bag-of-words similarity, choosing the top-100 tokens per relation as its context."?
>
> Answer: Context selection is the general term for passage selection/ranking. We compute TF-IDF-based similarity between the relation context and the passage. We choose top-100 tokens (based on their TF-IDF scores) as the relation context. We rephrased this in the revised paper.
>
> > Have you tried the comparison of the proposed method with GLRE + the passage ranking?
>
> Answer: Thanks for this suggestion. We conducted a new experiment on GLRE + our passage ranking, which has been added to Appendix D (Additional Experiments). The results show that, by using passage ranking to filter the inputs (for both train and test), GLRE is able to improve its precision and recall. However, this enhanced variant is still inferior to KnowFi by a substantial margin.

---

### Decision · Program_Chairs · 2021-08-18

**Decision:**

Accept

**Comment:**

This paper proposes a method for knowledge extraction from long documents, and evaluates it on a new dataset constructed for this purpose. All reviewers found the method novel, and the details added about the dataset and the model were appreciated. Thus I recommend acceptance.